# What Exactly Makes Age a Risk Factor for an Unfavorable Outcome after Mitral Valve Surgery?

**DOI:** 10.3390/jcm11236907

**Published:** 2022-11-23

**Authors:** Roya Ostovar, Filip Schröter, Ralf-Uwe Kühnel, Martin Hartrumpf, Johannes Maximilian Albes

**Affiliations:** Department of Cardiovascular Surgery, Heart Center Brandenburg, University Hospital Brandenburg Medical School, “Theodor Fontane”, Faculty of Health Sciences Brandenburg, 16321 Bernau, Germany

**Keywords:** outcome, aging, mitral valve surgery, frailty, kidney disease

## Abstract

Objective: Age has an undeniable impact on perioperative mortality. However, it is not necessarily a predictor of frailty per se, as older patients have different outcomes. To verify specific conditions underlying frailty, we examined demographics, comorbidities, frequency, and distribution of postoperative complications influencing outcomes in a challenging cohort of patients undergoing mitral valve surgery. Methods: The study enrolled 1627 patients who underwent mitral valve surgery. Patients younger than 40 years who had been diagnosed with endocarditis were excluded. Patients were divided into three groups with ages ranging from 40–59 (*n* = 319), 60–74 (*n* = 795), and >75 years (*n* = 513). Baseline, comorbidities, postoperative complications, and mortality were recorded. Results: The older the patients were, the more frequently they suffered from pre- and postoperative renal insufficiency (*p* < 0.001). The likelihood of postoperative renal failure requiring dialysis was significantly higher with pre-existing renal failure. There was a significant association between postoperative renal insufficiency and the development of postoperative pleural or pericardial effusion (*p* < 0.001, *p* = 0.016). A significant decrease in BMI was observed in patients >75 years of age compared to the 60–74 years group (27.3 vs. 28.2 kg/m^2^, *p* = 0.007). The development of critical illnesses such as myopathy and neuropathy (CIP/CIM) was age-dependent and increased significantly with age (*p* = 0.04). Hospitalization duration and mortality also increased significantly with age (*p* = 0.013, *p* < 0.001). Conclusions: It appears that elderly patients with advanced renal failure have a significantly higher risk of mortality, postoperative renal failure, need for dialysis, and possibly the development of pleural and pericardial effusions in mitral valve surgery. In addition, more frequent CIP/CIM with concomitant decrease in BMI in the most advanced age group indicate sarcopenia and thus an additional feature of frailty besides renal failure.

## 1. Background

Whereas aging per se is not a contraindication to cardiac surgery, performing cardiac surgery on elderly patients is particularly challenging. There is a big difference between a spry and a frail elderly patient. What exactly “spry” means and how to assess the outcome of elderly patients preoperatively is not clarified in the current literature. However, this topic has become more important in recent years. Not only is the population aging, but the number of cardiovascular diseases in elderly requiring surgery is increasing and shifting towards the more challenging procedures such as mitral valve surgery.

The proportion of elderly patients undergoing cardiac surgery is steadily increasing. While in Germany in 2012 13.8% of patients undergoing cardiac surgery were >80 years old, this was 20.7% in 2021. This in combination with a higher proportion of comorbidities and therefore a more complex perioperative risk profile is associated with higher mortality and poor outcomes despite more complex treatment.

In recent years, various tests have been investigated for practical assessment of frailty [1,2,3]. It is obvious that not every elder person is frail, but even elderly patients who appear spry at first glance are nevertheless at increased risk. Aging is a complex topic and has also been intensively investigated on a biomolecular level in recent years.

During aging, cardiac and non-cardiac organs change their structure and consequently lose crucial functional components such as the glomeruli of the kidney resulting in functional impairments that may well take place insidiously and unnoticed for a long period [4,5,6,7]. Various causes are discussed. One of them is oxidative stress [7]. The cardiovascular and renal systems are closely interconnected and influence each other, whereby this interlocking becomes increasingly problematic in older age due to the loss of function of the interdependent organs [8].

From clinical experience and other reports, we know that elderly patients more often suffer from chronic kidney failure and are more often affected by postoperative kidney failure.

The aim of this study was to analyze whether there is an age dependency of preoperative risk factors and perioperative complications and which of these factors are most significant. 

## 2. Patients and Methods

A positive ethics vote (E-02-20200923, dated 21 November 2020) was obtained prior to study initiation. Patient consent was waived in retrospective observational study design. After data collection and before statistical analysis, data were anonymized. A total of 1627 patients undergoing mitral valve surgery between 2009 and 2021 were included in the study. Exclusion criteria were age < 40 years and presence of endocarditis. The primary objective was identification of age-related risk factors and analysis of age-related outcome.

The primary endpoint was age-related outcome; secondary endpoints were age-related risk-factors and co-morbidities. 

### 2.1. Data Collection

In the present study, comorbidities were collected in addition to baseline and risk profile. Furthermore, perioperative diagnostic data including preoperative and postoperative echocardiographic findings, laboratory data, intraoperative course, and postoperative complications as well as hospitalization and hospital mortality were collected. Patients were divided into 3 age groups and compared with respect to risk factors and outcome. The young group included 319 patients between 40–59 years, the mid-age group included 795 patients between 60 and 74 years, and the elderly group included 513 patients ≥ 75 years. To analyze the impact of age on mitral valve surgery more closely, an additional subgroup analysis was performed in advanced age groups with subdivision into 70–75-year-olds of 330 patients, 75–80-year-olds of 381 patients, and 132 patients who were older than 80 years.

### 2.2. Statistical Analysis

Statistical analysis was performed using “R” [9]. To examine trends across the three age groups, the binary data were each tested using a Cochran–Armitage test. Thereafter, Holm–Bonferroni corrected Chi^2^ tests were performed as a post hoc test. For corresponding sites or for multiple choices (e.g., number of previous surgeries), a Kendall’s tau correlation was performed first, followed by Cochran–Armitage tests of the corresponding options. For baseline and risk profile, ANOVA and Kruskal Wallis, respectively, were applied first as a pretest, followed by Holm–Bonferroni corrected pairwise *t* tests for normally distributed data and Mann–Whitney U test for non-normally distributed data.

## 3. Results

### 3.1. Baseline and Comorbidities

The overall cohort was predominantly male (59.5%). However, with increasing age, the proportion of females within the groups increased significantly (21.32% vs. 40% vs. 53.2%, *p* < 0.001). The logistic EuroSCORE (Log. ES) to predict surgical mortality risk showed an average of 16.7% ± 18.6 in the entire cohort. Log. ES also increased significantly with increasing age (10.4% vs. 15.3% 22.7%, *p* < 0.001). Body mass index (BMI) was 27.8% ± 5.2 on average. We observed a significant BMI reduction between the mid-age group and the elderly group (28.2 vs. 27.3 kg/m^2^, *p* = 0.005). 

With echocardiography, we found no significant difference between the age groups regarding left ventricular ejection fraction. However, tricuspid annular plane systolic excursion (TAPSE) decreased, and pulmonary artery pressure increased significantly with age (*p* = 0.006 and *p* < 0.001). Laboratory N-terminal prohormone of brain natriuretic peptide (NT-ProBNP) as a sign of heart failure also showed no significant difference between the groups.

The presence of chronic renal failure increased significantly with age in all groups (*p* < 0.001). Within the groups, tumor disease (*p* = 0.003), atrial fibrillation (*p* < 0.001), arterial hypertension (*p* < 0.001), and coronary artery disease (*p* < 0.001) were observed significantly more often with increasing age. 

The proportion of peripheral arterial disease, chronic obstructive pulmonary disease, and pulmonary arterial hypertension increased with age in our cohort, but without statistically significant difference (Table 1).

### 3.2. Outcome

There was a significant association between postoperative development of renal failure and increasing age in all age groups (14.1%, 21.6%, and 28.1%, *p* < 0.001). Also, a correlation between aging and dialysis requirement was found (10.7%, 16.5%, 18.4%, *p* = 0.007). Furthermore, the proportion of pleural effusions and pericardial effusions was significantly higher with aging (*p* < 0.001, *p* = 0.016, respectively). 

It was interesting to observe a significant correlation between pleural effusion and preoperative renal insufficiency (*p* = 0.013). Postoperative kidney failure was significantly correlated with the frequency of pericardial effusion (*p* = 0.008) (Figure 1).

### 3.3. Age-Related Complications

Furthermore, we observed significantly more postoperative respiratory insufficiency with aging (*p* < 0.001). Also, we observed significantly more Critical Illness Myopathy and Critical Illness Polyneuropathy (CIP/CIM) with increasing age of the cohort (*p* = 0.049). Regarding age, we did not observe significant association between age groups and stroke rate, postoperative wound healing disorders, low output syndrome, postoperative pneumonia, urinary tract infection, systemic inflammatory response syndrome, pneumothorax, and cardiac arrhythmias (Table 2).

Hospitalization times increased with increasing age (*p* = 0.013). Furthermore, mortality increased with aging (*p* < 0.001). A significant correlation with mortality could be shown in female patients (*p* = 0.003), in patients with CHD (*p* = 0.003), and in patients with preoperative kidney insufficiency (*p* < 0.001). Furthermore, postoperative occurrence of kidney failure and dialysis dependency increased the mortality significantly (*p* < 0.001 and *p* < 0.001, respectively) (Figure 2). Postoperative development of CIP/CIM also significantly increased mortality (*p* = 0.01).

### 3.4. Subgroup Analysis in Advanced Age

The subgroup analysis of elderly patients showed very similar results. Thus, not only mortality was significantly higher with increasing age (*p* < 0.001), but also postoperative kidney failure (*p* = 0.009), postoperative pleural effusion (*p* = 0.006), and respiratory insufficiency (*p* = 0.009). 

Chronic renal insufficiency could be observed as a preoperative comorbidity with increasing age (*p* = 0.034). Furthermore, a trend towards more female patients (*p* = 0.017) and lower BMI (0.05) was found with increasing age (Table 3).

## 4. Discussion

In cardiac surgery, female gender is considered an additional mortality risk. EuroSCORE II calculates it as 0.22% points additional mortality risk. As women of advanced age are overrepresented in our cohort, the mortality risk is even higher. The increase in ES II with age was therefore predictable. This is not only due to age alone, but also to comorbidities that increase with age.

It is well known that aging and frailty are associated with sarcopenia [10,11,12,13]. The observed BMI decline in the older cohort was therefore of particular interest as it indicates muscle loss. 

Critical illness polyneuropathy, or CIP for short, is a frequently distally accentuated and symmetrical polyneuropathy that typically occurs in ventilated intensive care patients and can delay weaning. Critical illness myopathy is a myopathy that often occurs in patients receiving intensive care and makes weaning from the respirator difficult. The two are often observed together. The main clinical features are often a limp, symmetric weakness and reduction of deep tendon reflexes. In CIP, distal absence of pain and temperature sensitivity and involvement of phrenic nerves can be observed [14,15]. In some cases, in addition to peripheral, a disorder of the central nervous system, e.g., encephalopathy, becomes remarkable as a clinical feature [16]. Further clinical examination tests include evaluation of muscle strength according to the Medical Research Council [17]. Some patients in the intensive care unit may have inadequate clinical neurological examination, so in such cases, the electrophysiological diagnostic procedures such as electroneurography, electromyography, direct muscle stimulation, or measurement of serum creatinekinase are also used.

As CIP/CIM and respiratory failure were more common in the older cohort, it can be assumed that they are also caused by sarcopenia. LVEF and heart failure assessed by NT-proBNP did not differ between age groups. It is possible that older patients with impaired LVEF or significantly impaired cardiac function were less likely to be enrolled in the study, as older patients with poor LVEF are often denied surgery for fear of a poor outcome. On the other hand, it is quite possible that in the absence of coronary artery disease, ventricular function—at least the systolic component—is one of the bodily functions that deteriorates less with age than other systems such as the kidney or brain.

As expected, concomitant diseases increased significantly with age [18]. One of these was chronic renal insufficiency [19]. The risk of developing postoperative renal insufficiency requiring dialysis was also age-dependent. In addition, the significant increase in pleural and pericardial effusions with age found in this study may be at least partly due to renal insufficiency and dialysis dependence [19,20,21,22]. Although pleural effusion has various causes, including those related to inflammation, hypoproteinemia, or even malignancy, it is very likely that postoperative fluid overload associated with renal failure increased the rate of pleural and pericardial effusion.

The majority of patients recover fully after heart surgery without relevant complications and benefit from heart surgery. However, our results show that older patients have a higher risk, which is certainly not a novelty and very much in line with a large number of other publications. Postoperative complications, including length of hospital stay, and mortality were significantly increased in older patients. On the other hand, significantly more relevant comorbidities were observed in older patients. Of note, however, renal insufficiency and sarcopenia were particularly common in the elderly cohort and led to an unfavorable outcome. While renal failure is comprehensively addressed in the EuroSCORE system so that an outcome-prediction can be made, frailty itself remains elusive as only age alone is visible as a risk factor. Thus, not only a high BMI can be considered a risk factor, but also an unintentional BMI decline indicating frailty and thus the risk for an adverse outcome [23,24]. Therefore, in addition to the EuroSCORE, an assessment of frailty is needed to make an informed recommendation for surgical or interventional therapy in these patients on an individual basis, especially for demanding procedures with inherent risks such as mitral valve surgery.

An interesting point would be to compare the outcome and mortality after different surgical procedures in the elderly. This should be investigated in further studies.

### Limitation

This retrospective study has the typical limitation. In contrast to a prospective and carefully designed study, some data are missing, the procedures were carried out at the discretion of the individual surgeon, and the cohorts show inhomogeneities. In addition, the three age groups are somewhat arbitrarily chosen. 

A preoperative frailty test would be useful to analyze the influence of age and frailty on outcome after cardiac surgery, which could not be implemented in this study due to the retrospective design.

## 5. Conclusions

In an elderly cohort undergoing mitral valve surgery, meticulous assessment of the degree of renal function as well as a frail state as indicated by muscle loss is required. These patients need careful selection and planning that goes beyond the EuroSCORE assessment to verify the individual risks and calculate the extent to which surgical or interventional burden can be tolerated for procedures that themselves pose significant risk. “Less can then be more”. The older the patients are, the more one has to think thoroughly about what can or should be done.

## Figures and Tables

**Figure 1 jcm-11-06907-f001:**
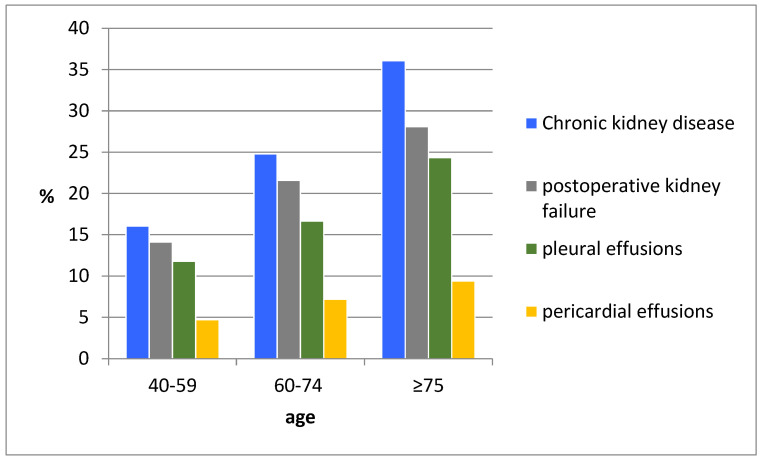
Age-related complication.

**Figure 2 jcm-11-06907-f002:**
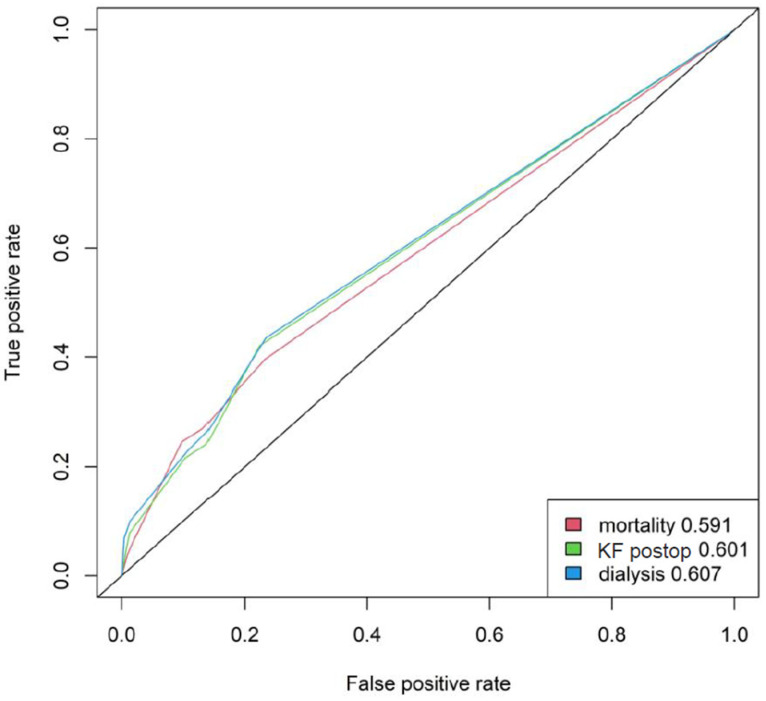
POC curve, KF postop: kidney failure postoperatively.

**Table 1 jcm-11-06907-t001:** Baseline and comorbidities.

Age-Distribution (Years)	40–59	60–74	≥75	*p*-Value
Gender (male)	78.7%	60%	46.8%	<0.001
Mean logistic EuroSCORE (%)	10.4 ± 15.1	15.3 ± 17.3	22.7 ± 20.6	<0.001
Body mass index (kg/m^2^)	27.7 ± 5.1	28.2 ± 5.5	27.3 ± 4.8	0.007
LVEF (%)	50.9 ± 13.6	51.3± 13.3	51.9 ± 12.5	n.s.
TAPSE (mm)	21.88 ± 5.88	20.95 ± 5.94	19.87 ± 5.22	0.006
PAP (mmHg)	34.1 ± 13.9	39.45 ± 16.63	41.01 ± 15.16	<0.001
NT-proBNP (pg/mL)	6445.5 ± 33,442.5	3904.2± 6918.5	4948.7 ± 19,704.3	n.s.
Chronic kidney disease	16.05%	24.79%	36.06%	<0.001
Stage 1	6.02%	9.94%	14.82%	
Stage 2	3.01%	3.50%	2.65%	
Stage 3	2.68%	9.24%	16.37%	
Stage 4	1.34%	0.84%	1.77%	
Stage 5	3.34%	1.26%	0.44%	
Coronary heart disease (CHD)	35.79%	54.41%	59.07%	<0.001
One vessel CHD	7.36%	14.13%	13.94%	
Two vessel CHD	5.69%	11.05%	13.27%	
Three vessel CHD	22.74%	28.95%	32.08%	
History of cancer	8.75%	11.66%	15.96%	0.003
Atrial fibrillation	25.59%	48.38%	55.28%	<0.001
Arterial hypertension	63.64%	83.64%	90.04%	<0.001
COPD	6.85%	11.37%	10.84%	n.s.
Peripheral arterial disease	4.38%	7.12%	6.84%	n.s.
Pulmonary arterial hypertension	13.7%	16.4%	14.9%	n.s.

LVEF: left ventricular ejection fraction, TAPSE: tricuspid annular plane systolic excursion, PAP: pulmonary artery pressure, NTproBNP: N-terminal prohormone of brain natriuretic peptide, CHD: coronary heart disease, COPD: chronic obstructive pulmonary disease.

**Table 2 jcm-11-06907-t002:** Postoperative complications.

Age Distribution	40–59	60–74	≥75	*p*-Value
renal failure	14.09% [42]	21.59% [152]	28.09% [125]	<0.001
dialysis requirement	10.74% [32]	16.45% [116]	18.43% [82]	0.007
pleural effusions	11.78% [35]	16.64% [118]	24.33% [109]	<0.001
pericardial effusions	4.7% [14]	7.19% [51]	9.4% [42]	0.016
respiratory insufficiency	12.79% [38]	17.63% [125]	23.66% [106]	<0.001
CIP/CIM	1.34% [4]	3.95% [28]	4.25% [19]	0.049
Stroke	1.69% [5]	3.16% [22]	3.41% [15]	0.204
Wound healing disorder	3.36% [10]	4.1% [29]	4.26% [19]	0.695
Low output syndrome	7.38% [22]	5.36% [38]	8.48% [38]	0.384
Pneumonia	6.04% [18]	7.62% [54]	9.17% [41]	0.114
urinary tract infection	1.01% [3]	1.27% [9]	2.47% [11]	0.093
SIRS	18.79% [56]	18.79% [133]	22.99% [103]	0.119
Pneumothorax	2.68% [8]	2.4% [17]	2.46% [11]	0.865
Atrial fibrillation	5.39% [16]	6.95% [49]	6.09% [27]	0.902
AV-Block II° or III°	2.36% [7]	2.27% [16]	2.26% [10]	0.998

CIP/CIM: Critical Illness Myopathy and Critical Illness Polyneuropathy; SIRS: systemic inflammatory response syndrome; AV-Block: atrioventricular block.

**Table 3 jcm-11-06907-t003:** Mortality, postoperative complications and comorbidities in advanced age.

Age-Distribution (Years)	70–75	75–80	>80	*p*-Value
Gender (female)	45.5%	52%	56.8%	0.017
Mean logistic EuroSCORE (%)	17.7 ± 18.8	21.4 ± 20.2	26.5 ± 21.4	<0.001
Body mass index (kg/m^2^)	27.6 ± 4.9	27.5 ± 4.9	26.5 ± 4.2	0.05
Chronic kidney disease	28.71%	35.71%	37.07%	0.034
Coronary heart disease	39.8%	40.48%	41.38%	n.s.
History of cancer	13.82%	14.33%	20.69%	n.s.
Atrial fibrillation	52%	51.2%	54.87%	n.s.
Arterial hypertension	85.2%	91.37%	86.21%	n.s.
COPD	13.13%	10.37%	12.17%	n.s.
Peripheral arterial disease	5.9%	6.82%	6.9%	n.s.
Pulmonary arterial hypertension	17.51%	14.63%	15.65%	n.s.
Postoperative Complications				
Acute kidney failure	24.5%	23.87%	40.35%	0.009
Dialysis requirement	19.4%	15.71%	26.32%	n.s.
Pleural effusions	17.88%	22.22%	30.43%	0.006
Pericardial effusions	5.96%	9.91%	7.89%	n.s.
Respiratory insufficiency	17.55%	21.62%	29.57%	0.009
CIP/CIM	3.64%	3.31%	6.96%	n.s.
Stroke	4.75%	2.44%	6.25%	n.s.
Low output syndrome	5.96%	7.51%	11.3%	n.s.
Pneumonia	7.28%	7.53%	13.91%	n.s.
SIRS	20.27%	21.02%	28.7%	n.s.
In-hospital Mortality	19.7%	22.57%	40.91%	<0.001

COPD: chronic obstructive pulmonary disease; CIP/CIM: Critical Illness Myopathy and Critical Illness Polyneuropathy; SIRS: systemic inflammatory response syndrome; n.s.: not significant.

## Data Availability

Data will not be published for privacy reasons and will be saved at the clinic.

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
