# Peer review of "What Exactly Makes Age a Risk Factor for an Unfavorable Outcome after Mitral Valve Surgery?"

_jcm, 2022, doi:10.3390/jcm11236907_

Round 1

Reviewer 1 Report

In this manuscript authors retrospectively analized impact of age and comorbidities on mitral valve surgery outcome. 

The paper confirms age is a major risk factor for cardiac surgery mortality and this data are already well known in literature. The manuscript is properly written, the overall structure is adequate to the aim; the introduction can be improved and should better explain correlation between age, frailty and surgery mortality and complications; the metholodogy and results are easy to understand, the discussion is clear but some statements has to be supported by adequate references.

Paper has some major limitations:

- to give a better picture of impact of age on mitral valve surgery, a suddivision in advanced age stages with data on mortality and comorbidities should be added (e.g. 70-75 years, 75-80, 80-85 years, > 80 years)

- a frailty scale, to stratify the risk of elderly patients, should be used

- a comparison with risk of other cardiac surgery in elderly could offer a clearer point of view about mitral valve surgery mortality

- a definition of CIP/CIM and how you have diagnosed it should be inserted

Author Response

Responses to the comments of Reviewers

Dear Reviewers,

We would like to thank you for careful and thorough reading of this manuscript and for the thoughtful comments. The suggested corrections were made. So we have sent the revised manuscript, and a version containing all the changes to be visible.

Reviewer 1: The paper confirms age is a major risk factor for cardiac surgery mortality and this data are already well known in literature. The manuscript is properly written, the overall structure is adequate to the aim; the introduction can be improved and should better explain correlation between age, frailty and surgery mortality and complications; the metholodogy and results are easy to understand, the discussion is clear

We have now added some references to the manuscript and supplemented the introduction. All changes are highlighted in yellow

Comment 1:  to give a better picture of impact of age on mitral valve surgery, a suddivision in advanced age stages with data on mortality and comorbidities should be added (e.g. 70-75 years, 75-80, 80-85 years, > 80 years)

Answer 1: Thank you for comment. A subgroup analysis to more precisely assess the impact of age seems to be beneficial. So we performed it and added corresponding results including table under section 3.4. A supplementary part in method can also be found. Despite small differences regarding comorbidities, interestingly there seem to be many similarities regarding outcome. Especially the mortality differences are very pronounced.

Change 1: Methods, Data collection, page 2 ; results, subgroup analysis, pase 5 and 6

Comment 2: Afrailty scale, to stratify the risk of elderly patients, should be used

Answer 2: We agree with the reviewer. A frailty scale based on the various frailty tests (Hand Grip, Timed Up and Go, Mini Mental test and etc.) would of course be very useful and important to investigate the influence of age on outcome. Unfortunately, none of these tests were originally performed in the patients and we could not integrate them into the study due to the retrospective design. We will certainly take this into account for future prospective studies. Accordingly, we mentioned this in the limitation of the study.

Change 2: Limitation, page 7

Comment 3: a comparison with risk of other cardiac surgery in elderly could offer a clearer point of view about mitral valve surgery mortality.

Answer3:  Mortality after mitral valve surgery compared to the other surgical procedures e.g. myocardial revascularization via bypass surgery in elderly patients is really an interesting point. We also believe that there could be relevant differences in terms of outcome and mortality as well as risk profile and comorbidities.

The problem is that in the ethics vote of the present study, the included patients are only patients after mitral valve surgery. This inhibits us from collecting or evaluating patient data after other types of surgery.

Nevertheless, it is an important and interesting point, so we mentioned it in discussion. We will also analyze these in more detail in the next study. Thank you for the food for thought.

Change 3: discussion, page 7

Comment 4: a definition of CIP/CIM and how you have diagnosed it should be inserted

Answer 4: The definition and diagnostic procedures of CIP/CIM were added.

Change 4: page 6

Reviewer 2 Report

The authors aimed to verify specific conditions underlying frailty and influencing outcomes in elder patients undergoing mitral valve surgery. The main findings suggest that elderly patients with advanced renal failure have a significantly higher risk of mortality, postoperative renal failure, need for dialysis, and possibly the development of pleural and pericardial effusions in mitral valve surgery. Importantly, the authors suggest these patients to be more carefully selected, as the outcomes go beyond the standard EuroSCORE assessment. The paper warrants the basis for important discussion, how much surgical or interventional burden can be tolerated for the high risk procedures in the elder patients. 

Author Response

Responses to the comments of Reviewers

Dear Reviewers,

We would like to thank you for careful and thorough reading of this manuscript and for the thoughtful comments. The suggested corrections were made. So we have sent the revised manuscript, and a version containing all the changes to be visible.